# Human and Bacterial Toll-Interleukin Receptor Domains Exhibit Distinct Dynamic Features and Functions

**DOI:** 10.3390/molecules27144494

**Published:** 2022-07-14

**Authors:** Eunjeong Lee, Jasmina S. Redzic, Travis Nemkov, Anthony J. Saviola, Monika Dzieciatkowska, Kirk C. Hansen, Angelo D’Alessandro, Charles Dinarello, Elan Z. Eisenmesser

**Affiliations:** 1Department of Biochemistry and Molecular Genetics, School of Medicine, University of Colorado Anschutz Medical Campus, 12801 E 17th Ave, Aurora, CO 80045, USA; eunjeong.lee@cuanschutz.edu (E.L.); jasmina.redzic@cuanschutz.edu (J.S.R.); travis.nemkov@cuanschutz.edu (T.N.); anthony.saviola@cuanschutz.edu (A.J.S.); monika.dzieciatkowska@cuanschutz.edu (M.D.); kirk.hansen@cuanschutz.edu (K.C.H.); angelo.dalessandro@cuanschutz.edu (A.D.); 2Department of Medicine, School of Medicine, University of Colorado Anschutz Medical Campus, 12801 E 17th Ave, Aurora, CO 80045, USA; charles.dinarello@cuanschutz.edu; 3Department of Internal Medicine, Radboud University Medical Center, 6525 GA Nijmegen, The Netherlands

**Keywords:** toll-like receptor, toll-interleukin receptor domain, NAD

## Abstract

Toll-interleukin receptor (TIR) domains have emerged as critical players involved in innate immune signaling in humans but are also expressed as potential virulence factors within multiple pathogenic bacteria. However, there has been a shortage of structural studies aimed at elucidating atomic resolution details with respect to their interactions, potentially owing to their dynamic nature. Here, we used a combination of biophysical and biochemical studies to reveal the dynamic behavior and functional interactions of a panel of both bacterial TIR-containing proteins and mammalian receptor TIR domains. Regarding dynamics, all three bacterial TIR domains studied here exhibited an inherent exchange that led to severe resonance line-broadening, revealing their intrinsic dynamic nature on the intermediate NMR timescale. In contrast, the three mammalian TIR domains studied here exhibited a range in terms of their dynamic exchange that spans multiple timescales. Functionally, only the bacterial TIR domains were catalytic towards the cleavage of NAD^+^, despite the conservation of the catalytic nucleophile on human TIR domains. Our development of NMR-based catalytic assays allowed us to further identify differences in product formation for gram-positive versus gram-negative bacterial TIR domains. Differences in oligomeric interactions were also revealed, whereby bacterial TIR domains self-associated solely through their attached coil-coil domains, in contrast to the mammalian TIR domains that formed homodimers and heterodimers through reactive cysteines. Finally, we provide the first atomic-resolution studies of a bacterial coil-coil domain and provide the first atomic model of the TIR domain from a human anti-inflammatory IL-1R8 protein that undergoes a slow inherent exchange.

## 1. Introduction

While the discovery of toll-like receptors (TLRs) and interleukin-1 receptors (IL-1Rs) has revolutionized our understanding of signaling in innate immunity and inflammation [1,2,3], there remains a surprising lack of details that connect the initial signaling events on the outside of the cell to their intracellular interactions. Both receptor families share a common intracellular domain, called the toll-interleukin receptor (TIR) domain (Figure 1A), which are proposed to recruit intracellular adaptor TIR domains through multiple interactions [4]. However, the TIR interactome has almost exclusively been extrapolated from structures of individual TIR domains and computational modeling of their interactions [5,6,7,8,9,10]. This means that experimentally determined atomic models of TIR complexes remain absent. These TIR domains are also found within many bacteria, where they are thought to target mammalian TIR complexes to interrupt the endogenous signaling within their host [11,12,13]. However, structural elucidations of bacterial/mammalian TIR complexes have yet to be characterized. Such a lack of experimentally determined TIR complexes is surprising considering their critical roles in innate immunity and infection. The inability to isolate specific TIR domain complexes has been proposed to result from an inherent cooperative feature that induces large assemblies only under specific conditions and which is called signaling by cooperative assembly formation (SCAF) [14]. The formation of such assemblies is likely reliant on the inherent plasticity of its constituents, which allows for their rapid conformational changes to occur during signaling. Thus, we conducted a comprehensive study to evaluate the solution behavior of bacterial and human TIR domains to address whether these domains are intrinsically dynamic and to address potential differences in their functional interactions.

TIR-containing domains are often attached within the same polypeptide to domains that induce oligomeric interactions and may be primed to form further assemblies upon various triggers. For example, upon ligand stimulation, mammalian TIR-containing receptors oligomerize to their extracellular domains, as described for TLR1/TLR2 [15] (Figure 1A), while there are cytosolic “adaptor” mammalian TIR-containing proteins that form intricate assemblies through their attached domains [16]. These adaptor assemblies include the death domain of Myd88, known as the myddosome, which self-associates to form larger complexes with the death domains of kinases [17], and octameric assemblies of human SARM1 mediated by its SAM domain [18,19,20,21]. Interestingly, bacterial TIR-containing proteins also self-associate through a separate domain, which is an N-terminally attached dimerization domain thought to form a coil-coil (CC) (Figure 1B). However, these bacterial dimerization domains bear no similarity in terms of sequence to any protein structure previously solved. As for assemblies mediated directly by TIR domains, a subset of mammalian TIR domains have been shown to self-associate in the absence of additional oligomeric domains, which may be more generally due to the oxidative state of cysteine residues. For example, the mammalian adapter MAL forms fibrils in a temperature-dependent manner that coincides with the breaking of an intramolecular disulfide bond [22], and the TLR4-TIR domain is proposed to form homodimers through the formation of an intermolecular disulfide of a conserved cysteine present within many mammalian TIR domains [23] (Figure 1C). In fact, X-ray crystal structures of TIR domains from TLR1, TLR2, TLR6, and TLR10 indicate that this same conserved cysteine resides at a crystallographic dimer interface and forms an observable intermolecular disulfide within TLR1 and TLR6 TIR domains [6,24]. However, beyond TLR4, it is currently unknown as to whether this conserved cysteine facilitates direct homo- and heterodimerization.

Bacterial TIR domains are found within pathogenic- and antibiotic-resistant bacteria that include *Acinetobacter baumannii* [25] and *Staphylococcus aureus* [13,26,27,28]. In fact, bacterial TIR-containing proteins are emerging as virulence factors [13,26,28], which is exemplified by the TIR domain encoded within methicillin-resistant *Staphylococcus aureus* (MRSA) strains responsible for recent waves of “flesh eating” infections [29]. Several bacterial TIR domains have been shown to be enzymes that comprise NADase activity, which also appear to be common in plants [19,20,21,30]. However, one mammalian TIR-containing protein, SARM1 (sterile alpha and TIR motif containing 1), has also been shown to comprise NADase activity that is involved in axon degeneration [30,31]. Catalysis of NAD^+^ is mediated by a relatively well conserved glutamic acid also present in many mammalian TIR domains (Figure 1C), suggesting that these mammalian counterparts could also harbor NADase activity.

In this study, we applied a combination of biochemical and biophysical methods to a panel of both bacterial and mammalian TIR domains to determine whether they are inherently flexible, undergo self-association, and cleave small molecules such as NAD^+^. For the bacterial TIR-containing proteins, we produced three bacterial TIR-containing proteins along with their isolated CC and TIR domains (Appendix A). These included a *Staphylococcus aureus* TIR protein (tirS) recently shown to harbor NADase activity [31], an uncharacterized *Acinetobacter junii* TIR protein (tirA), and an uncharacterized *Enterococcus* TIR protein (tirE). For the mammalian TIR domains, we produced three isolated TIR domains, which included those from TLR1 (TLR1-TIR), TLR2 (TLR2-TIR), and IL1-R8 (IL-1R8-TIR). TLR1/TLR2 is a well-known proinflammatory receptor complex [6,15], and IL-1R8 is proposed to “break” inflammation by interfering with its signaling [32]. The studies conducted here revealed that these TIR domains broadly exhibit inherent dynamics but exhibit distinctly different interactions that include, in the case of only the bacterial TIR domains, NADase activity and, in the case of the mammalian TIR domains, multiple cysteine-mediated oligomeric interactions. Such studies reveal the intrinsic dynamic nature of these TIR domains and provide a rationale for why structural elucidations of their complexes have been so challenging.

## 2. Results

### 2.1. Bacterial Proteins of tirS, tirA, and tirE All Cleave NAD^+^ and Related Coenzymes

As tirS along with several other TIR-containing bacterial proteins have recently been shown to be enzymatic in that they cleave NAD^+^ and related coenzymes [31], we sought to determine whether tirA and tirE are catalytic and whether mammalian TIR domains may exhibit a similar activity. Considering the known structural homology between TLR1-TIR and TLR2-TIR domains in terms of their X-ray crystal structures [6] and that they both conserve glutamic acid (Figure 1C), we used TLR2-TIR to probe the activity of a mammalian TIR. Additionally, while the bacterial tirS served as a positive control, based on its recently identified catalytic activity [31], we hypothesized that the mammalian IL-1R8-TIR domain would serve as a negative control, as it does not comprise the same catalytic glutamic acid.

Full-length bacterial TIR-containing proteins and mammalian TIR domains were incubated in protein-depleted red blood cell extracts to broadly identify small molecules that incurred changes through mass spectrometry (MS) analysis. Incubations of all three bacterial TIR-containing proteins (tirS, tirA, and tirE) resulted in diminishments to NAD^+^, NADP^+^, NADH, and NADPH (Figure 2A, top). However, no changes to these coenzymes were observed upon incubation with the human TIR domains (Figure 2A, bottom). Although catalyzed product differences were not identified within these complex mixtures, adenosine was increased within the bacterial TIR incubations, which is suggestive of in-source fragmentation for the direct products. Thus, despite the presence of catalytic glutamic acid in TLR2-TIR, only the bacterial TIR-containing proteins are catalytic towards the coenzymes.

We used NMR to further interrogate the explicit catalytic activities of the bacterial TIR-containing proteins purified here by incubating tirS, tirA, and tirE with NAD^+^ and followed by catalysis in real time (Figure 2B). The catalytic efficiencies of these bacterial NADases were significantly different, as tirS was capable of turning over all of the NAD^+^ within 30 min at only 1 μM, while both tirA and tirE required over an order of magnitude higher concentrations. Comparisons of their product spectra with commercially available standards of NAD^+^, ADP-ribose, and nicotinamide clearly illustrated that both gram-positive bacterial TIR proteins (tirS and tirE) cleave NAD^+^ into both ADP-ribose and nicotinamide (Figure 2C). In contrast, the single gram-negative TIR protein used here (tirA) exhibited a distinctly unique resonance at 8.62 ppm, with a concomitant loss of its ADP-ribose resonances at 8.22 ppm (Figure 2C, arrow). This is consistent with a modified ADP-ribose previously proposed to be cyclic ADP-ribose (cADP-ribose) produced by a different TIR-containing protein from *Acinetobacter baumannii* [31].

The direct detection of TIR-mediated catalysis of NAD^+^ via 1D-NMR provides a powerful means to monitor activity in real time and probe the effects of products on catalysis. We had initially anticipated that this assay could be used to monitor Michaelis–Menten kinetics to provide comparisons between TIR domains. However, higher concentrations of NAD^+^ resulted in lower initial velocities, indicating that these TIR domains did not subscribe to standard Michaelis–Menten kinetics. To understand why catalytic turnover was not increased within increased enzyme concentrations, we specifically focused on tirE. While the isolated TIR domain of tirE was not amenable to the recombinant purification described further in the next section, we generated a chimera of GB1-tirE-TIR to probe its activity in the absence of its CC domain, whereby GB1 is the Streptococcal protein G immunoglobulin B1 domain tag that increases solubilization. Incubation of the products with either GB1-tirE-TIR (Appendix A) or the full-length tirE (Appendix A) led to changes in the initial velocities, which indicates a slowing of the turnover for nicotinamide and an increase in turnover for ADP-ribose (Appendix A). Thus, our data indicate that cellular concentrations of products provide an additional layer of control for TIR-mediated catalysis and that the cleaved NAD^+^ products are different between these gram-negative and gram-positive bacterial TIR domains. Whether this is a general trend will need to be confirmed through further investigations into other bacterial TIR domains.

### 2.2. Bacterial TIR Domains of tirS, tirA, and tirE Undergo Chemical Exchange with Dimerization Modulated by Their Coil-Coil Domains

We initially focused on the isolated bacterial TIR domains alone, as NMR solution studies of bacterial TIR domains have only been performed on the TIR-containing protein from *Yersinia pestis.* Specifically, these previous studies revealed highly line-broadened resonances for the *Yersinia pestis* TIR indicative of micro-millisecond (μs-ms) dynamics [11]. Such exchange precluded assignments and solution structural studies of the *Yersinia pestis* protein, begging the question of whether other bacterial TIR domains exhibit such an inherent exchange. However, initial bacterial expression studies failed for the isolated TIR domains of tirS and tirA but were possible for the Glu→Ala mutants that removed the catalytic glutamic acid residues (Appendix A). As TIR domains that include the tirS domain were previously expressed in a bacteria-free system [31], these findings could indicate that recombinant bacterial expression of the catalytically active wild type (WT) bacterial TIR domains here leads to cell death, with the subsequent selection of those cells exhibiting low expression. The recombinant purification of both ^15^N-labeled tirS-TIR E216A and tirA-TIR E206A resulted in relatively low yields, although the latter did produce enough protein to collect an ^15^N-HSQC (Figure 3A). Interestingly, only a subset of resonances within the tirA-TIR E206A spectrum that usually correlate with disordered regions were visible, suggesting that the folded regions undergo an exchange like that of the previously characterized *Yersinia pestis* TIR [11].

Based on the difficulty of detecting the resonances of tirA-TIR E206A, we chose a different strategy for the tirE-TIR domain that was employed for the *Yersinia pestis* TIR [11]. Specifically, the small ~50-residue GB1 tag was N-terminally linked to facilitate a comparison to GB1 alone. The chimeric construct of GB1-tirE-TIR was readily purified along with that of the GB1 control (Appendix A). However, almost no new resonances were observed for the GB1-tirE-TIR relative to the GB1 control spectrum (Figure 3B, gray versus red). In fact, upon incubation of the NAD^+^ that we discovered is cleaved by GB1-tirE-TIR, as described above, little to no chemical shifts are observed. This suggests that the exchange of tirE-TIR still leads to a loss in resonances and that most of the resonances that include the regions responsible for substrate binding are still not observable when bound to the NAD^+^ cleavage products. Neither temperature, pH, nor different buffers (HEPES and MES) resolved this absence of resonances. Thus, the previous observation of an inherent chemical exchange for the *Yersinia pestis* TIR appears to be a wider trait of all the bacterial TIR domains studied here. While the underlying reasons that lead to chemical exchange could be due to internal motions or the formation of larger oligomeric interactions, the analytical methods described herein suggest these bacterial TIR domains alone do not self-associate.

To address whether the bacterial TIR domains may be forming larger oligomeric complexes in solution that could underlie their loss in resonances, analytical sizing was used. A subset of bacterial TIR domains that includes the TIR domain from *Paracoccus denitrificans* (PnTIR) [33] and the TIR-containing protein from *Brucella melitensis* (TcpB) have been amenable to structural studies using X-ray crystallography [34,35]. While both these previously studied bacterial TIR domains exhibit homodimerization, the TcpB TIR also exhibits evidence of larger oligomeric species [35]. Thus, tirS, tirA, and tirE along with their subdomains were all subjected to size-exclusion chromatography (Figure 3C–E). The full-length bacterial proteins and isolated CC domains migrated to a more significant degree than the TIR domains, which is consistent with the CC domain, but not the TIR domain, inducing dimerization. From a quantitative assessment of their elution behavior, both the full-length proteins and their CC domains migrated as dimers as predicted, while the isolated TIR constructs migrated as monomers (Appendix A). Thus, the bacterial CC domains studied here are responsible for the dimeric nature of the proteins, while the TIR domains themselves are monomeric. We noted that although weak self-association under the higher concentrations used for NMR could contribute to chemical exchange and the subsequent absence of observed resonances, the higher yield of GB1-tirE-TIR afforded its analytical sizing at concentrations similar to those used in NMR, with no concentration-dependent changes in migration. We therefore concluded that while weak self-association cannot completely be ruled out to have a contribution to chemical exchange, it is likely that there is an inherent dynamic process (or processes) that underlies this exchange for multiple bacterial TIR domains.

### 2.3. The CC Domain Is Not Completely Helical

As there are currently no experimentally solved structures of the CC domain from any bacterial TIR-containing protein, which makes predictive models unreliable, experiments were sought here to provide insight into their structure. While the electrostatic nature of CC domains resulted in highly soluble proteins (pI~9), crystal screens proved unsuccessful in our hands, which may also be due to significantly disordered regions. Thus, we turned to NMR solution studies, whereby ^15^N-HSQC spectra of CC domains were initially collected for tirS, tirA, and tirE (Figure 4A–C). Resonance dispersion varied for these CC domains, which exhibited very different dependencies based on sample conditions. This suggests that the folded CC domains undergo exchange broadening, although not to the extent of the above described TIR domains, which result in the complete absence of their resonances under similar conditions. For example, the tirS-CC spectrum was well-dispersed under high salt conditions (Figure 4A), while only a subset of resonances of tirA-CC were observed in both low and high salt (Figure 4B). In contrast, good dispersion was observed for tirE-CC for both high and low salt conditions (Figure 4C). Considering the high salt necessary for tirS-CC and the poor dispersion, with only a subset of resonances observable, of tirA-CC, we pursued NMR studies of tirE-CC.

Assignment strategies that included both CO and side chain CA, CB resonances were successful in identifying ~80% of the backbone resonances within the tirE-CC domain (Figure 4D). However, even with deuteration, several stretches were not able to be identified, which included residues 35–38, 42–49, and 53–68, which culminated in 31 missing amides of the 144 amides that comprise the tirE-CC domain (146 residues when including its two prolines). These missing resonances could be due to exchange or could be due to the particularly slow tumbling of the elongated dimer proposed for such CC domains. Chemical shifts were used to calculate the secondary structure propensities for tirE-CC. Surprisingly, these indicated that residues 102–146 do not sample an a-helical structure and instead exhibit a β-strand-like structure that is predicted to be a random coil by CSI version 2.0 [36] (Figure 4D). Thus, these data provide the first experimental descriptions of the secondary structure of a bacterial CC domain and indicate that the entirety of at least tirE-CC is not helical. This weak β-strand propensity for residues 102–146 may allow for an extended region and subsequent flexibility for the attached TIR domains of these bacterial proteins (Figure 4E). Unfortunately, full structural calculations were hampered due to the complete lack of NOEs, which was likely due to the elongated structural fold of the dimer that leads to anisotropic tumbling. Considering that no structural insight to any bacterial CC domain exists, which is exemplified by the inability of even AlphaFold to predict the monomeric structure of tirE-CC (https://www.uniprot.org/uniprotkb/A0A097BYB4/entry, accessed on 30 May 2022), we pursued a structural model based on our NMR chemical shifts. Specifically, backbone chemical shifts were used in Chemical Shift (CS)-Rosetta [38], which produced the expected two α-helices and allowed for a prediction of the electrostatic surface of this region (Figure 4F). While it is still difficult to predict the dimerization surface, our findings here suggest that tirE-CC structural manifold necessary for dimerization lies within this region, with extended linkers of residues 102–146 connecting the C-terminal TIR domain.

### 2.4. TLR1 and TLR2 TIR Domains form Disulfide-Linked Homodimers

A conserved cysteine residue within an interaction surface responsible for self-association has been identified in the structures of several mammalian TIR domains solved to date, including TLR1 (C707) [6], TLR2 (C713) [6], TLR6 (C712) [24], and TLR10 (C706) [5]. This is illustrated here for the previously solved structure of TLR1-TIR along with the conserved sequences of this region of other mammalian TLR TIR domains and IL-1R8, which all comprise this conserved cysteine (Figure 5A,B). Upon purifying both the TLR1-TIR and TLR2-TIR domains, we observed homodimers to readily form even in the presence of reducing agents and that they could be minimized in the presence of 50 mM DTT or completed diminished by the mutation of this conserved cysteine residue (Figure 5C,D, top). These point mutations are still well-folded, as indicated by their ^15^N-HSQC spectra (Appendix A). Thus, homodimerization is mediated by this same conserved cysteine in TLR4, which can be pharmacologically targeted to suppress TLR4 signaling [23]. However, TLR4 is the only TLR that is currently known to signal as a homodimer, suggesting that the observed tendency to form homodimers within the TLR1-TIR and TLR2-TIR domains observed here may serve to block their formation of the competent signaling complex. In general, this mechanism of disulfide-bond dimerization within the reducing environment of the cell is referred to as a “dock-and-lock” mechanism [39,40,41], as it comprises a relatively weak interaction referred to as “docking” that may be followed by the covalent formation of a transient disulfide bond, with this being referred to as ”locking”.

### 2.5. TLR1 and TLR2 TIR Domains Form Disulfide-Linked Heterodimers

We next sought to evaluate whether a potential disulfide-mediated heterodimerization occurs for the mammalian TIR domains through a combination of NMR, pull-down assays, and crosslinking mass spectrometry (CL-MS) (Figure 6). As NMR is particularly sensitive to weak interactions, NMR was used to probe the “docking” interaction between TLR1-TIR and TLR2-TIR domains. Specifically, the highly dispersed ^15^N-labeled TLR2-TIR spectrum was monitored with the addition of unlabeled TLR1-TIR (Figure 6A, left), suggesting the presence of a weak interaction illustrated by the binding isotherm of several resonances that were estimated to be within the millimolar range (Figure 6A, right). A similar titration of the full-length tirA served as a negative control (Figure 6B), also indicating that this bacterial TIR domain does not block TLR2 function by direct engagement, as previously proposed for other bacterial TIR domains [42]. To obtain evidence of “locking”, we first utilized a GST-TLR1-TIR chimera for incubations with untagged TIR domains and subsequent GST affinity pull-downs. GST-TLR1-TIR pull-downs indicated interactions with untagged TLR1-TIR, TLR2-TIR, and IL1-R8-TIR (Figure 6C, lanes 1, 3, and 5, respectively), while a GST control did not pull down TIR domains under identical conditions (Appendix A, lanes 1 and 3). Interestingly, these interactions were not solely reliant on the conserved cysteine within the interfaces of TLR1-TIR and TLR2-TIR within their respective X-ray crystal structures, as their point mutations were still pulled down (Figure 6C, lanes 2 and 4, TLR1-TIR C707A and TLR2-TIR C713A). Such biochemical findings are consistent with the recent discovery that a second cysteine, TLR1 C667, also contributes to the TLR1/TLR2 response proposed to be mediated by Zn and is also the only other conserved cysteine within mammalian TLR TIR domains [43]. However, we could not further explore the potential role of Zn in inducing heterodimer formation, as Zn often induces protein aggregation and induces an interaction with the control GST alone that is likely non-specific aggregation (Appendix A, lanes 2 and 4).

We used CL-MS to identify which cysteine residues are responsible for TLR1-TIR/TLR2-TIR disulfide-mediated heterodimer formation. Samples of 6H-tagged TLR1-TIR and 6H-tagged TLR2-TIR domains were prepared in a similar fashion to that of the GST-TLR1 sample preparations featuring DTT. Samples were split for either no further reduction or further reduction with 5 mM tris(2-carboxyethyl)phosphin) (TCEP) prior to trypsin digestion for subsequent mass spectrometry analysis, and these groups were referred to as “Pre-Treatment” and “Post-Treatment”, respectively (Appendix A). Four heterophilic cross-links were identified between TLR1-TIR and TLR2-TIR domains that are consistent with the ability of these two domains to cross-link in the GST-TLR1 pull-downs upon mutations of TLR1-TLR C707A and TLR2-TIR C713A (Figure 6D). We note that although TLR1-TIR comprises three cysteine residues (C667, C686, and C707) and TLR2-TIR comprise four cysteine residues (C640, C673, C713, and C750), disulfide-linked peptides were not identified for one such cysteine within each domain that included TLR1-TIR C707 and TLR2-TIR C750. Specifically, trypsin cleavage for TLR1-TIR C707 likely resulted in relatively long peptides that were not amenable to mass spectrometry analysis, and TLR2-TIR C750 may simply be unreactive compared to the other cysteine residues. Homophilic cross-links were also identified for TLR1-TIR (Figure 6E) and TLR2-TIR (Figure 6F). Although no peptide could be identified for TLR1-TIR C707, such homophilic cross-links were identified for TLR2-TIR C713, which showed resistance to further treatment of DTT. At least two cross-links likely included internal disulfide bonds, such as TLR1-TIR C667–C686 and TLR1-TIR C640–C673, that are in close proximity within the TLR1-TIR domain and were well-protected post-treatment with TCEP.

Collectively, these studies indicate that cysteine residues within TLR TIR domains are reactive and modulate homodimer and heterodimer formation even in the presence of reducing conditions. However, as opposed to homodimerization, which appears to be specific for one cysteine within each TLR TIR domain (TLR1-TIR C707 and TLR2-TIR C713, see Figure 5), heterodimerization is relatively non-specific and comprises most of the cysteine residues (see Figure 6). Considering the preponderance of recent studies illustrating how cysteine chemistry is coupled to conformational changes within other TIR domains [22,23,44], these findings could suggest that TLR signaling at the cellular membrane may also be modulated by disulfide-bond chemistry.

### 2.6. Mammalian Pro-inflammatory TIR Domains Exhibit a Range of Chemical Exchange

We next questioned whether a dynamic exchange is present within mammalian TIR domains of TLR1 and TLR2, as they are in the bacterial TIR domains described above which result in severe line-broadening. A recent solution study has shown that the TLR1-TIR domain exhibits a global inherent dynamic [43], which resulted in severe line-broadening for many residues, which is consistent with our observations here (Appendix A). Specifically, several regions of TLR1-TIR exhibited high R2 relaxation rates, several resonances were unobservable likely due to exchange, and most of the protein exhibited low order parameters. This means that most of the TLR1-TIR domain is flexible and exhibits motions on multiple timescales. Here, we extended studies to the TLR2-TIR domain to determine whether this TIR also exhibits an inherent dynamic.

**Figure 7 molecules-27-04494-f007:**
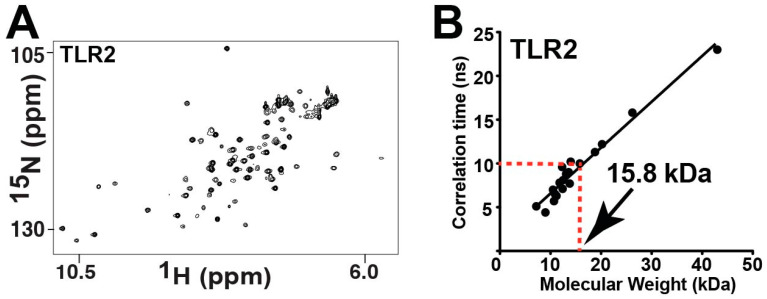
Chemical exchange within TLR2-TIR leads to the loss of numerous resonances. (**A**) ^15^N-HSQC of 460 μM TLR2-TIR. (**B**) A correlation time of 22 ± 1.5 ns for TLR2-TIR was determined from the ratio of 51 R2/R1 relaxation rates and a molecular weight of 15.8 kDa estimated based on a comparison to previously measured correlation times [45].

It was immediately obvious that the TLR2-TIR domain also undergoes exchange, as less than half of the expected resonances are observed within its ^15^N-HSQC spectrum (Figure 7A). As opposed to the TLR1-TIR C707A mutation, which induced only minor changes to its spectrum (Appendix A), the TLR2-TIR C713A mutation led to the emergence of many new resonances (Appendix A). This indicates that there is a global dynamic coupling in TLR2-TIR to this site and its mutation likely shifts an equilibrium that leads to slightly less line-broadening. Unfortunately, only marginal differences are observed to the spectrum upon increasing the temperature (Appendix A) or through the use of TROSY-based methods (Appendix A). In order to determine whether the lack of resonances of TLR2-TIR is an inherent dynamic of the monomer or due to chemical exchange mediated by self-association, we first utilized NMR relaxation to estimate the correlation time of TLR2-TIR (Appendix A). By measuring R2 and R1 relaxation rates, the average R2/R1 ratio was determined to be 22.18, which was then used to calculate a correlation time of 10.2 ns using the previously described equation [46]. This correlation time can then be used to estimate the molecular weight using protein standards, as we have previously described [45]. Using this correlation time for TLR2-TIR and a plot of protein standards (Figure 7B), the associated molecular weight was estimated at 15.8 kDa, which is slightly less than the calculated molecular weight based on the sequence of 18.2 kDa. Thus, our data suggests that TLR2-TIR is largely monomeric and that the lack of observable resonances is likely due to an inherent dynamic that leads to severe line-broadening for the majority of the domain. In other words, unlike the apparent extremes of bacterial TIR domains, which are completely line-broadened (Figure 3A,B), and TLR1-TIR, which results in the majority of its resonances being partially line-broadened [43], TLR2-TIR lies in between, with over half of its resonances being line-broadened due to an inherent exchange.

### 2.7. The IL-1R8-TIR Domain Undergoes a Slow Exchange

While we hypothesized that all TIR domains are inherently dynamic, we were particularly interested in addressing whether this was true for the IL-1R8-TIR domain, which functions differently from the pro-inflammatory TLR1 and TLR2 proteins. IL-1R8 is unique in that it is anti-inflammatory and is referred to as a “break on inflammation” [32], which would allow us to assess whether an anti-inflammatory TIR domain is also intrinsically dynamic. IL-1R8 comprises only a single extracellular Ig-like domain, which is proposed to be the innate receptor for IL-37 [47,48], yet no atomic-resolution studies have been performed on this receptor that includes its intracellular TIR domain.

We first addressed the solution behavior of the IL-1R8-TIR domain in regard to both its hydrodynamics and spectral resolution. As the IL-1R8-TIR domain does contain the conserved TLR “WC” sequence (Figure 5B), we probed whether it forms disulfide-linked dimers similar to the TLR1-TIR and TLR2-TIR domains. Only one monomer was observed by size-exclusion chromatography, which indicates homodimers are not readily formed (Figure 8A, top). However, as the IL-1R8-TIR domain gives rise to an asymmetrical elution profile unlike that of the distinct dimer/monomer peaks of the TLR1 and TLR2 TIR domains (Figure 8A, middle and bottom), this does suggest that weak self-association may persist. We next sought to determine whether the IL-1R8-TIR domain lacks resonances due to a dynamic exchange, as found for TLR2-TIR above and recently found for TLR1-TIR [43]. In contrast to the other mammalian TIR domains, the spectrum of the IL-1R8-TIR domain of residues 160–310 does not appear to be significantly line-broadened and comprised a sufficient number of resonances that could account for the entire protein (Figure 8B). We therefore sought to characterize the structure and dynamics of this module to determine whether a TIR domain with anti-inflammatory activity is also inherently dynamic.

While the spectral resolution was indicative of a well-folded protein (Figure 8B), our elucidation of the backbone assignments allowed us to identify the existence of multiple resonances for the same amides, which revealed a relatively slow exchange process (Figure 8B, inset). The majority of these duplicate resonances were obvious, as there were two resonances adjacent to each other with similar CA and CB chemical shifts. Beyond confirming the presence of a slow inherent dynamic within IL-1R8-TIR, such assignments also facilitated secondary structure propensities, as shown here by the experimental chemical shift deviations of the average assigned CA positions from their random coil values (Figure 8C). Positive values indicate α-helical propensities and negative values indicate β-strand propensities [36]. However, despite the use of deuteration to maximize signal intensities within standard backbone assignment experiments, there were two contiguous stretches of residues, 44–49 and 73–80, that were unable to be identified. These missing resonances include the conserved “WC” found in TLR1 and TLR2, which is IL-1R8 W77 and C78 (corresponding to W237 and C238 within the full-length IL-1R8). Relaxation studies identified the elevated R2 relaxation rates of neighboring residues to these missing stretches (Figure 8D and Appendix A), such as A52 and residues 81–86, which is consistent with the possibility that such neighboring residues are under fast exchange while the missing stretches undergo exchange on the slow timescale. These may still be detecting the same physical exchange process, as exchange is dependent on the chemical shift differences of sampled populations that may simply be much larger for the missing stretches of residues 44–49 and 73–80. Interestingly, the same adjacent residues to these regions are also highly dynamic within the ps–ns timescale, as indicated by their increased R1 relaxation rates (Figure 8E and Appendix A), illustrating a range of dynamics for these regions.

The lack of resonances for multiple regions of IL-1R8-TIR makes a full solution structure analysis difficult, yet our backbone assignments do allow for a low-resolution structure model using CS-Rosetta predictions. Specifically, CS-Rosetta utilizes fragment libraries based on chemical shift assignments in order to build structural ensembles, which has been successfully used for proteins much larger than TIR domains [38]. Here, the five most probable solutions from CS-Rosetta calculations are shown, which are consistent with several observations (Figure 7F). First, a superposition of the most probable structural ensemble solutions indicates that both missing regions of residues 44–49 and 73–80 are adjacent to each other. This lends evidence to the coupled dynamic exchange process of these regions, which leads to their severe line-broadening. Second, the prediction also suggests that those residues that define the “BB-loop” in TIR domains corresponds to an α-helix in IL-1R8-TIR of residues 53–56. Such helical loops corresponding to this region are not uncommon, as they are also observed within adaptor TIR domains such as MAL [22] and Myd88 [12]. In fact, a DALI search of this IL-1R8-TIR structure indicates that MAL both alone and within the context of its filamentous form exhibit the highest structural homology (Appendix A). This is particularly interesting, considering that IL-1R8-TIR and MAL share almost no sequence similarities other than the conserved “WC” (residues 156–157 within MAL). Third, the conserved IL-1R8-TIR C78 is pointed inwards, just as it is within MAL C157 (Appendix A red arrow), and is therefore not exposed as it is for TLR1-TIR and TLR2-TIR within their X-ray crystal structures (Figure 7G compared to Figure 5A). This is consistent with biochemical findings that indicate a cysteine-mediated homodimer does not readily form as observed for TLR1 and TLR2. Finally, the regions that exhibit multiple resonances are also largely clustered to both each other and those amides for which resonances are absent (Figure 7). This could suggest that these regions are collectively involved in a globally coupled conformational exchange that facilitates their inherently slow dynamics. The underlying reason of this exchange could be due to any number of reasons that include the slow proline isomerization of either one or multiple proline residues (P15, P27, P46, P50, P96, P108, P111, P130, P135, and P148), although this may be difficult to assess at this point considering that the wild type protein is marginally unstable (glycerol was found to stabilize the wild type protein for NMR studies, as described in the Materials and Methods section). Nonetheless, we can conclude that while the IL-1R8-TIR domain is a dynamic module, its inherent dynamic comprises a very slow exchange process that leads to the appearance of multiple resonances.

This inherently slow exchange discovered within IL-1R8-TIR could underlie conformational selection that allows for the sampling of important conformations responsible for binding to a cognate partner, such as TLR1-TIR. For example, it is known that IL-1R8 blocks inflammation for a variety of TLRs that include TLR1 [49], which could be through a direct interaction between these two TIR domains. As IL-1R8-TIR was found to form a disulfide-linked heterodimer with the GST-TLR1 above (Figure 6C), this suggested that IL-1R8-TIR works through a dock-and-lock mechanism. Only minor CSPs were detected for ^15^N-labeled IL-1R8-TIR in the presence of either TLR1-TIR and TLR2-TIR (Appendix A), which indicates that the initial binding events may be weaker than the high micromolar concentrations used for the independently purified TIR domains. Analogous to the disulfide-linked TLR1-TIR/TLR2-TIR heterodimer complex, the heterodimer complex of GST-TLR1-TIR/IL-1R8-TIR could not be scaled up. Such studies indicate that a signaling complex comprises other stabilizing factors such as cytosolic adaptor TIR domains known to be downstream of TLR signaling [8,14,16,22]. Thus, the IL-1R8-TIR is unique in its anti-inflammatory function. Its intrinsic dynamics are also unique, in that it is the only TIR domain studied here that results in a slow exchange on the NMR timescale.

## 3. Discussion

We identified a range in the inherent dynamics of TIR domains and revealed distinct differences in the interactions that occur between bacterial and mammalian TIR proteins. Dynamically, in contrast to the bacterial TIR domains that undergo a global exchange, which leads to their extreme line-broadening, the human TIR domains probed here exhibited a range in terms of chemical exchange. Specifically, most of the resonances for TLR1-TIR observed here that is consistent with a recent study reporting exchange on the fast timescale for this domain [43]. In contrast, both TLR2-TIR and IL-1R8-TIR exhibit slower exchange processes. For TLR2-TIR, over half of the protein undergoes an exchange that leads to the disappearance of resonances, which is similar to case of the bacterial TIR domains, while the IL-1R8-TIR domain exhibits a slow exchange process that leads to the disappearance of several contiguous regions, with the simultaneous appearance of multiple resonances for others. Functionally, only bacterial TIR domains are catalytic towards NAD^+^ and dimerize solely through their CC domains, while the human TIR domains can form both homo- and hetero-dimers through their cysteine residues that are not present within their bacterial counterparts. Finally, the fact that all the TIR domains studied here are intrinsically dynamic does support a signaling mechanisms reliant on the sampling of conformations that would facilitate large-scale cooperative changes proposed by a SCAF model [14].

In the case of the bacterial TIR domains, while chemical exchange leads to extreme line-broadening within all three of the TIR domains studied here, NMR-based activity assays showed both similarities and differences. For example, tirS, tirA, and tirE all cleave NAD^+^, yet tirA cleavage resulted in a different product, which is likely cADP-ribose. This may have wide implications, as cyclic ADP-ribose has been shown to modulate cellular signaling in T cells [50] and may therefore implicate a further control of some bacterial TIR domains on the host cell. Our findings therefore begin to pinpoint specific functions that these bacterial TIR domains have in modulating the host cell metabolome and potentially signaling even in the absence of direct interactions with mammalian TIR domains. As for the CC domain that lies N-terminal to these bacterial TIR domains, all three of the domains studied here were also extremely line-broadened. However, we were able to assign much of the NMR solution resonances of the tirE-CC domain, providing the first atomic-resolution study of any such domain. Chemical shift propensities indicate that this tirE-CC domain is not entirely helical in solution, as previously proposed [42]. While such a helical structure was based on plant CC [51], this may not be too surprising in retrospect based on the low sequence similarity.

For mammalian TIR domains, only weak intermolecular interactions were detected by NMR, while biochemical studies detected the formation of cysteine-dependent homodimers and heterodimers, which suggests a role for thiol biochemistry. Such a mechanism has been described for other signaling complexes as a “dock-and-lock” mechanism [39,40,41], which comprises the initial weak “docking” followed by a disulfide linked “locking”. Both TLR1-TIR and TLR2-TIR readily form homodimers through their conserved cysteine residues (TLR1 C707 and TLR2 C713, respectively), which is the same conserved cysteine responsible for TLR4 homodimerization, which comprises the LPS receptor [23]. Considering that TLR4 signals as a homodimer and the heterodimization of TLR1 to TLR2 is required for signaling, this could suggest that TLR1 and TLR2 homodimerization acts as a break on inflammatory signaling. In other words, during inflammatory responses where TLRs such as TLR1 are highly expressed on immune cells that include natural killer cells [52], self-association may serve as a means to limit inflammatory responses that may become harmful for the cell. As opposed to the high specificity of TLR1-TIR and TLR2-TIR dimerization that was reliant on a single conserved cysteine, heterodimerization was less specific, with nearly all possible disulfide-linkages identified by MS analysis. However, it is possible that differential cysteine-mediated heterodimization may underlie pro-inflammatory signaling or allow for initiating larger oligomeric complexes such as those found within MAL filaments [22]. In contrast to TLR1 and TLR2, our low-resolution structure of IL-1R8-TIR suggests that this cysteine is pointing inwards and may therefore explain why this TIR does not readily homodimerize as the TLR1-TIR and TLR2-TIR domains do. Caution should be taken in this interpretation, as this region within IL-1R8-TIR is also not detected in solution and was therefore modeled by CS-Rosetta without any experimental chemical shift restraints. In fact, this region most definitely undergoes a dynamic exchange that could expose the same conserved cysteine to form intermolecular cross-links and suppress TLR signaling. To this point, GST-TLR1-TIR does pull down IL-1R8-TIR in addition to TLR2-TIR under moderate reducing conditions, which suggests that heterodimer formation may also be disulfide-mediated. While complexes of the TLR1-TIR and TLR2-TIR domains have eluded structural determination, our mass spectrometry data suggests that these domains form heterogenous disulfide-mediated heterodimers and homodimers, likely hampering such structural studies. This is consistent with a recent report on TLR1 signaling that indicates that the conserved C673 is important for its pro-inflammatory activity [43].

In general, the findings here are also consistent with the emerging roles of disulfide chemistry in mammalian TIR signaling. For example, a combination of X-ray crystallographic studies [7], solution NMR studies [44], and high-resolution cryo-EM studies [22] have collectively revealed that the cytosolic adaptor TIR domain, MAL (otherwise known as TIRAP), undergoes drastic conformational changes associated with changes to intramolecular disulfides. Specifically, two disulfides observed within the crystal structure of MAL, C142–C174 and C89–C134, become reduced in solution, and this may underlie the filamentous formation that provides for a competent signaling complex. Glutathione modifications of MAL are also observed, which modulate its interactions with Myd88 [7], adding further post-translational control in terms of signaling. Disulfide biochemistry modulates global structure and activity even on the outside of the cell, such as in the cases of intramolecular disulfides that modulate homodimer interactions in IL-8 [53] and the thioredoxin-mediated reduction of a disulfide that inactivates IL-4 [54]. Understanding the crucial missing intermediate steps that link receptor interactions on the outside of the cell to downstream signaling mediated by TIR domains and determining how bacterial TIR domains block these events is still an on-going effort. The internal dynamics and disulfide-mediated mechanisms identified here may provide connections as to how TLR signaling responds to oxidative stress [55], whereby intermolecular disulfides may facilitate heterodimers that drive pro-inflammatory signaling while simultaneously producing homodimers that abrogate signaling under conditions of excess receptor expression [56,57,58,59,60,61]. Considering the wealth of therapeutic interventions that have emerged from structural studies of the initiating receptor complexes on the outside of the cell [62], our elucidations of TIR interactions may provide exciting new avenues to therapeutically block for multiple diseases integrally tied to inflammation.

## 4. Materials and Methods

### 4.1. Annotated Proteins and Expression Plasmids

All engineered constructs correspond to publicly available sequences and sub-domains derived from these. *Staphylococcus aureus* TIR protein (tirS) corresponds to NCBI accession sequence WP_000114516.1. *Acinetobacter junii* TIR protein (tirA) corresponds to GenBank accession sequence RTE47094.1. *Enterococcus* TIR protein (tirE) corresponds to both *Enterococcus* strains and an *Enterococcus* phage with NCBI accession sequences YP_009103093.1 and WP_102970086.1, respectively. The separately engineered bacterial CC and TIR domain sequences are also listed in Appendix A, with the noted exception that tirE-TIR was attached to a his-tagged streptococcal G protein (GB1). GB1 had the following sequence: MAKYYHHHHHHQYKLILNGKTLKGETTTEAVDAATAEKVFKQYANDNGVDGEWTYDDATKTFTVTEGG. Human TLR1-TIR corresponds to NCBI accession sequence NP_003254.2, whereby the entire intracellular TIR region of residues 625–786 was used in this study. Human TLR2-TIR corresponds to NCBI accession sequence NP_001305716.1, whereby the entire intracellular TIR region of residues 633–784 was used in this study. Human IL-1R8-TIR corresponds to accession AAH03591, whereby residues 160–310 were used in this study. All proteins were cloned and expressed in either in pET21b or pJ401k as previously described [63,64,65]. Specifically, while most proteins described in this study were expressed with pET21b, both GB1 and the GB1-tirE-TIR utilized pJ401k that expressed with higher yields. All p21b constructs comprised a 6xHis-tag followed by a thrombin cleavage site.

### 4.2. Protein Expression and Purification

Unlabeled proteins were typically grown in 4 L of Luria Broth and labeled proteins in 4 L of LB and media swapped via low-speed centrifugation to 1 L of M9 minimal media (12 g/L Na_2_HPO_4_, 6 g/L KH_2_PO_4_, 1 g/L NaCl, 1 g/L NH_4_Cl, 2 g/L glucose, 2 mL of 1 M MgSO_4_ 100 mL of 1 M NaCl CaCl_2_, 10 mg/L thiamine). For pET21b or pJ401k, media was supplemented with 100 mg/mL ampicillin or 50 mg/mL kanamycin, and cells were induced at an OD (600) of 0.6. M9 was either ^15^N-labeled ammonium chloride for ^15^N-labeled protein or ^15^N-labeled ammonium chloride, ^13^C-labeled glucose, and grown in deuterium for ^2^H,^15^N,^13^C-labeled tirE-CC. The latter was utilized for subsequent NMR assignments of tirE-CC. Briefly, for purifications, all proteins were homogenized via sonication with “denaturation buffer“ (5 M guanidine, 50 mM Na_2_HPO_4_, pH 7.5, 500 mM NaCl, 10 mM imidazole) and applied to a Ni-affinity resin column (Sigma, St. Louis, MO, USA). Elutions were dialyzed against “refold buffer” (100 mM tris, pH 7.5, 100 mM NaCl) supplemented with 1 M arginine and then dialyzed against refold buffer alone. Precipitate was removed and the proteins were concentrated for the subsequent removal of the 6xHis-tag using thrombin (Sigma, St. Louis, MO, USA). Finally, mammalian TIR domains were further purified via Superose-12 and bacterial TIR domains further purified via a Superdex-75 in “NMR buffer” (50 mM Na_2_HPO_4_, pH 6.5, 150 mM NaCl). All mammalian TIR domains were also supplemented with 1 mM DTT, which was added upon refolding in denaturation buffer, refold buffer, and NMR buffer. We note that ^15^N-HSQC spectra of TLR1-TIR and TLR2-TIR match those previously reported [9,43].

### 4.3. Nuclear Magnetic Resonance

NMR samples were prepared in the above defined NMR buffer, and all 1D and 2D spectra presented here were collected on a Varian 900 MHz spectrometer (Agilent Technologies, Pasadena, CA, USA) equipped with a cryo-probe at 25 °C, unless otherwise described. As IL-1R8-TIR was stable for only several hours, 10% glycerol was added to increase the lifespan of each sample to several weeks as monitored through ^15^N-HSQC spectra. For 1D spectra that followed NADase activities, first the indicated concentrations of NAD^+^ and products were incubated, and the specified TIR-containing proteins were added prior to collection, with a total of 16 scans averaged over each minute. For relaxation data, standard Varian BioPack R1 and R2 relaxation experiments were collected on a Varian 900 MHz spectrometer. Assignment experiments for ^2^H,^13^C,^15^N-labeled tirE-CC were collected on a Bruker 800 equipped with a cryo-probe at 25 °C, which included an HNCACB, HNcoCACB, HNCO, and HNCACO. Assignment experiments for ^2^H,^13^C,^15^N-labeled IL1-R8-TIR were collected on a Varian 600 equipped with a cryo-probe at 30 °C, which included an HNCACB, HNcoCACB, HNCO, and HNCACO. Assignments for tirE-CC and IL1-R8-TIR have been deposited in the Biological Magnetic Resonance Bank with the accession codes 51474 and 51473, respectively.

### 4.4. Cross-Linking and GST Pull-Down Experiments

GST pull-down experiments were performed with 200 mL slurries of GST beads (Sigma, St. Louis, MO, USA) using mini-spin columns (Micro Bio-Spin, BIO-RAD, Hercules, CA, USA). Prior to loading GST columns, either 250 μM GST-TLR1 or GST alone was incubated with 300 μM of untagged TIR domains (20% excess) in 10 mM DTT to reduce all potential disulfides. After complexes were incubated for 10–15 min, they were concentrated via centrifugation (Amicon Ultra 0.5 mL Centrifugal Filter, MilliporeSigma, Berlington, MA, USA) and serially diluted with no DTT buffer to approximately 200 μM DTT. Samples were incubated overnight at 4 °C prior to loading onto GST columns and eluted with 30 mL of buffer with 30 mM glutathione (Sigma, St. Louis, MO, USA). Control experiments with GST alone were performed the same way, and those in the presence of Zn utilized incubations of 200 μM Zn.

### 4.5. Mass Spectrometry

To analyze the metabolites consumed by TIR domains in red blood cell extracts, ultra-high-performance liquid chromatography–mass spectrometry (UHPLC–MS) was used. Analyses were performed as previously published [66]. Briefly, the analytical platform employs a Vanquish UHPLC system (Thermo Fisher Scientific, San Jose, CA, USA) coupled online to a Q Exactive mass spectrometer (Thermo Fisher Scientific, San Jose, CA, USA). Samples were resolved over a Kinetex C18 column, 2.1 × 150 mm, with a particle size of 1.7 µm (Phenomenex, Torrance, CA, USA) equipped with a guard column (SecurityGuard^TM^ Ultracartridge—UHPLC C18 for 2.1 mm ID Columns—AJO-8782—Phenomenex, Torrance, CA, USA) using an aqueous phase (A) of water and 0.1% formic acid and a mobile phase (B) of acetonitrile and 0.1% formic acid for positive ion polarity mode and an aqueous phase (A) of water:acetonitrile (95:5) with 1 mM ammonium acetate and a mobile phase (B) of acetonitrile:water (95:5) with 1 mM ammonium acetate for negative ion polarity mode. Samples were eluted from the column using either an isocratic elution of 5% B flowed at 250 µL/min and 25 °C or a gradient from 5% to 95% B over 1 min, followed by an isocratic hold at 95% B for 2 min, flowed at 400 µL/min and 45 °C. The Q Exactive mass spectrometer (Thermo Fisher Scientific, San Jose, CA, USA) was operated independently in positive or negative ion mode, scanning in Full MS mode (2 μscans) from 60 to 900 *m*/*z* at a resolution of 70,000, with 4 kV spray voltage, 45 shealth gas, 15 auxiliary gas. Calibration was performed prior to analysis using the PierceTM Positive and Negative Ion Calibration Solutions (Thermo Fisher Scientific). Acquired data was then converted from .raw to .mzXML file format using Mass Matrix (Cleveland, OH, USA). Samples were analyzed in a randomized order, with a technical mixture injected after every 15 samples to qualify instrument performance. Metabolite assignments, isotopologue distributions, and correction for expected natural abundances of deuterium, 13C, and 15N isotopes were performed using MAVEN (Princeton, NJ, USA) [67].

For proteomic analysis of TLR1-TIR and TLR2-TIR disulfides, 90 μM 6H-tagged TLR1-TIR was added to 110 μM 6H-tagged TLR2-TIR (~20% excess of TLR2) in 1 mM DTT for 10–15 min with a similar treatment as to that described above for mixtures of gel samples to a final 100 μM DTT. For proteomic analysis of TLR1-TIR and TLR2-TIR disulfides, 90 μM 6H-tagged TLR1-TIR was added to 110 μM 6H-tagged TLR2-TIR (~20% excess of TLR2) in 1 mM DTT for 10–15 min with a similar treatment as that described above for mixtures of gel samples to a final 100 μM DTT. Precipitation of TLR1-TIR and TLR2-TIR was performed by the addition of trichloroacetic acid to a final concentration of 25% and proceeded for 30 min at 4 °C. The proteins were washed twice with ice-cold acetone (−20 °C) and subsequently solubilized in 8 M urea/0.1 M Tris (pH 8.5). The sample was then equally divided into two parts, one of which was alkylated with 50 mM 2-chloroacetamide for 15 min at room temperature in the dark, while the other was reduced with 5 mM tris(2-carboxyethyl)phosphine) at room temperature for 20 min prior to alkylation. Both samples were diluted with four volumes of 100 mM Tris-HCl (pH 8.5) and digested with 1µg of sequencing grade trypsin (Promega) overnight at 37 °C. Following the overnight digestion, formic acid (FA) was added to the samples to a final concentration of 1%, and tryptic peptides were cleaned with Pierce^TM^ C_18_ tips (ThermoFisher Scientific^TM^) following the manufacturer’s protocol. Digests were dried in a vacuum centrifuge and resuspended in 0.1% FA in mass spectrometry-grade water and subjected to liquid chromatography tandem mass spectrometry (LC-MS/MS) using an Easy nLC 1000 instrument coupled to a Q-Exactive HF mass spectrometer (both from ThermoFisher Scientific). Digested peptides were loaded on a C18 column (100 μM inner diameter × 20 cm) packed in-house with 2.7 μm Cortecs C18 resin and separated at a flow rate of 0.4 μL/min with solution A (0.1% FA) and solution B (0.1% FA in ACN) and under the following conditions: isocratic at 4% B for 3 min, followed by 4–32% B for 102 min, 32–55% B for 5 min, 55–95% B for 1 min, and isocratic at 95% B for 9 min. MS/MS was performed using a data-dependent acquisition (DDA) top 15 method with the dynamic exclusion set to 20s. Fragmentation spectra were searched against the amino acid sequences of TLR1-TIR and TLR2-TIR using the MSFragger-based FragPipe computational platform [68]. The precursor-ion mass tolerance and fragment-ion mass tolerance were set to 10 ppm and 0.2 Da, respectively, and carbamidomethylation (C) and oxidation (M) were set as variable modifications. Disulfide crosslinks were identified using the pLink (v2.3.10, Bioinformatics group, Chinese Academy of Sciences, Beijing, China) software package. Parameters used for the search include “disulfide bond (HCD-SS)” crosslink, a fixed modification of C-1[C] for free reduced Cys, oxidation of Met, and a trypsin cleavage specificity with up to three missed cleavages. Peptide masses between 600 and 6000 with a peptide length of 6 to 60 amino acids were considered. Precursor and fragment tolerances were set to 20 ppm, with further filtering for +/−10 ppm and FDR filtering at <5% at the PSM level.

## 5. Conclusions

Toll-Interleukin-Receptor (TIR) domains are present within all kingdoms of life that include mammals, where they serve as signaling modules involved in immunity, and bacteria, where they serve as virulence factors that modulate the host cell. Despite this importance of TIR domains in mammalian signaling and bacterial infection, a specific understanding of how they interact with themselves, with specific metabolites, and with each other remains largely unknown. By studying and comparing three bacterial and three human TIR domains here, we have revealed fundamental differences between these sub-families. Specifically, only bacterial TIR domains serve as enzymes to cleave host NAD and self-associate solely through their previously uncharacterized dimerization domain, referred to as a coil-coil domain. Conversely, mammalian TIR domains utilize disulfide biochemistry to form both homodimers and heterodimers, which is consistent with recent findings that have revealed intimate roles between the redox state of the cell and signaling. In general, TIR signaling has been proposed to rely on unknown triggers that induce largescale conformational changes, called signaling by cooperative assembly formation, which predicts inherently dynamic modules confirmed here for all TIR domains. The inherent dynamics of these proteins leads to difficulties in monitoring all resonances due to severe line-broadening and has hampered high-resolution solution structure determination. Despite this, solution studies here have still provided the first atomic-resolution description of the secondary structure of any bacterial dimerization coil-coil domain and the first to describe the structure of the human IL-1R8-TIR domain involved in modulating innate immune signaling.

## Figures and Tables

**Figure 1 molecules-27-04494-f001:**
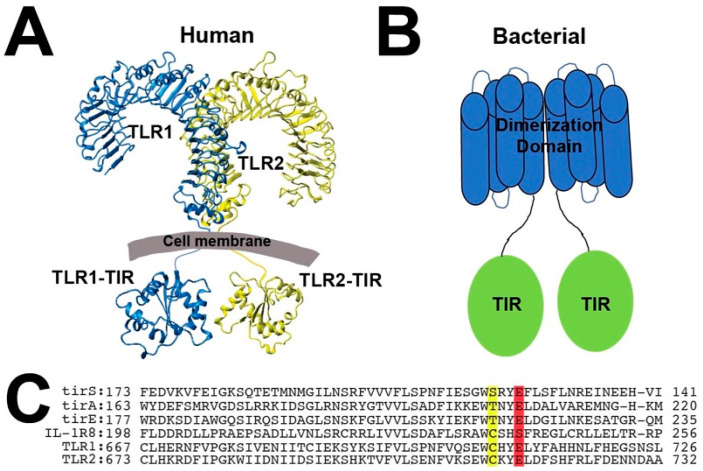
Secondary structure of and sequence comparison of bacterial and mammalian TIR-containing proteins. (**A**) The mammalian TLR1/2 model that includes the crystal structure of the extracellular complex, PDB accession number 2Z7X [15], and the individually crystallized intracellular TIR domains, PDB accession numbers 2FYV and 1FYW [6]. (**B**) Cartoon representation of bacterial TIR-containing proteins that comprise a CC dimerization domain (blue) and TIR domain (green). (**C**) Sequence comparison of the three bacterial TIR-containing proteins (tirS, tirA, and tirE) and three mammalian TIR domains (IL-1R8, TLR1, and TLR2), illustrating the conserved mammalian cysteine (yellow) and bacterial catalytic glutamic acid (red).

**Figure 2 molecules-27-04494-f002:**
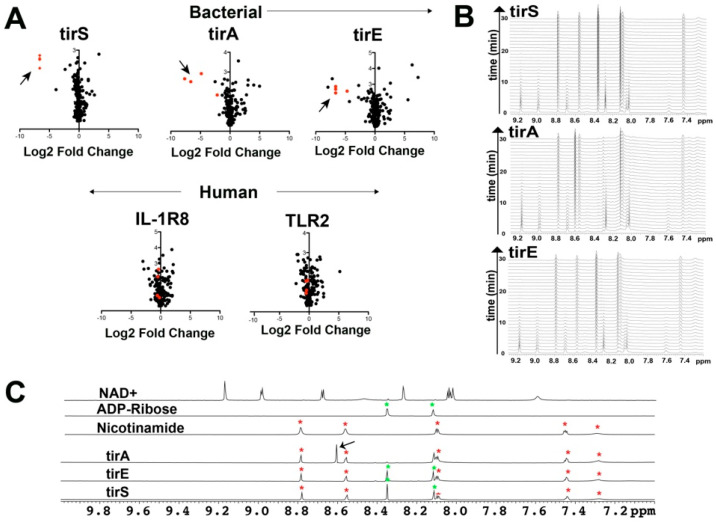
Characterizing TIR-containing catalytic activity. (**A**) Volcano plots derived from MS analysis are shown for 24 h incubations of 1 μM bacterial TIR proteins that include tirS, tirA, and tirE and mammalian TIR domains from IL-1R8 and TLR2 incubated in protein-depleted RBC extracts relative to a buffer control. Arrows delineate the reduction in NAD^+^, NADP^+^, NADH, and NADPH (red). (**B**) The time course of one-dimensional NMR spectra is shown for 1 mM NAD^+^ after incubation with 1 μM tirS, 43 μM tirA, and 25 μM tirE. (**C**) 1D-spectra of NAD^+^, ADP-ribose, and nicotinamide along with the product spectra of NAD^+^ incubated with tirS, tirA, and tirE. Resonances from ADP-ribose and nicotinamide are denoted green and red, respectively, while the arrow denotes the single difference within tirA.

**Figure 3 molecules-27-04494-f003:**
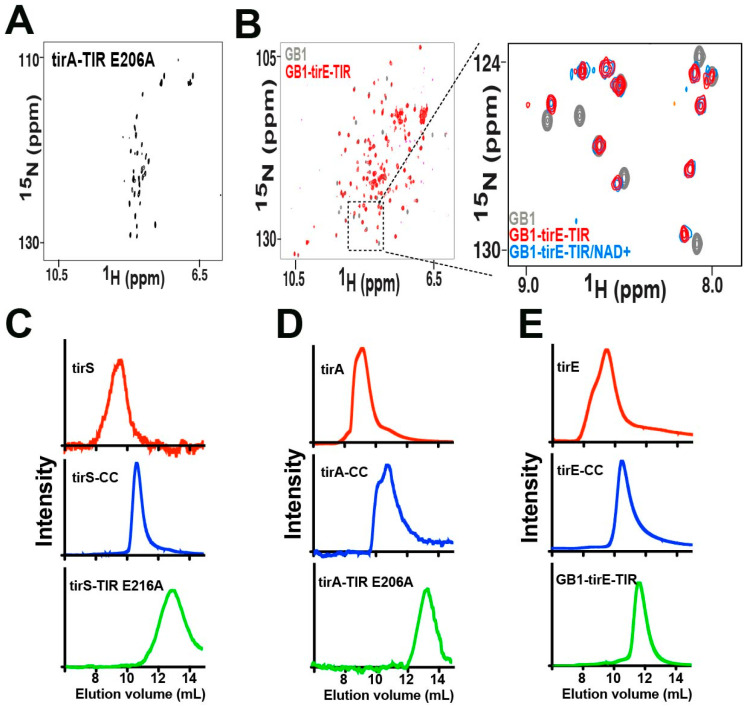
Biophysical and biochemical analysis of bacterial TIR-containing proteins. (**A**) The ^15^N-HSQC spectra of the tirA-TIR E206A mutant. (**B**) The ^15^N-HSQC spectra of GB1 alone (gray) and GB1-tirE-TIR (red) with an enlargement including the additional spectrum of GB1-tirE-TIR incubated with saturating concentrations of NAD^+^ (blue). (**C**) Analytical sizing of full-length tirS (red), tirS-CC (blue), and the tirS-TIR E216A mutant (green). (**D**) Analytical sizing of full-length tirA (red), tirA-CC (blue), and the tirA-TIR E206A mutant (green). (**E**) Analytical sizing of full-length tirE (red), tirE-CC (blue) and GB1-tirE-TIR (green). Sample loads were all 100 mL of 200 μM, except tirS-TIR E216A and tirS-TIR E206A, which were 20 μM and 50 μM, respectively.

**Figure 4 molecules-27-04494-f004:**
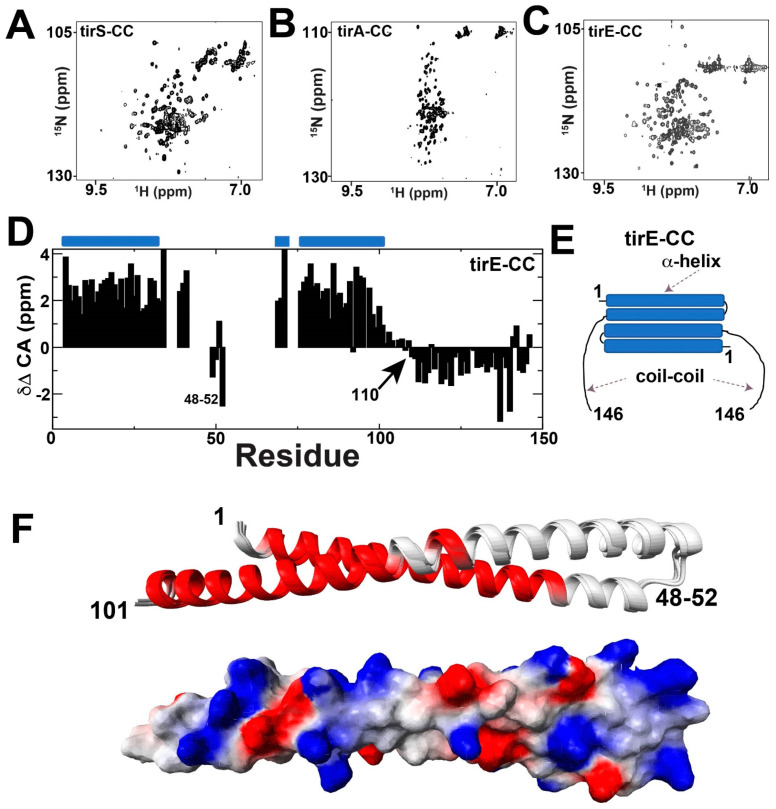
NMR studies of TIR-containing CC domains. (**A**) tirS CC in 50 mM Na_2_HPO_4_, 500 mM NaCl, pH 6.5. (**B**) tirA CC in 50 mM Na_2_HPO_4_, 50 mM NaCl, pH 6.5. (**C**) tirE CC in 50 mM Na_2_HPO_4_, 150 mM NaCl, pH 6.5. (**D**) CA chemical shift propensities (δΔ CA) of tirE-CC were calculated by the CA chemical shift for each residue type and subtracting random coil CA resonance reported for that same residue type at the BMRB (http://www.bmrb.wisc.edu/, accessed on 30 May 2022). Secondary structure predictions of helices (blue barrel) are shown, as predicted by CSI version 2.0 [36]. (**E**) Proposed secondary structure based on the chemical shift propensities of tirE-CC. (**F**) Top five structures determined from CS-Rosetta with α-helices predicted from experimental chemical shifts (red) and electrostatic surface representation (red is negative charge, blue is positive charge), shown below as calculated in Molmol [37].

**Figure 5 molecules-27-04494-f005:**
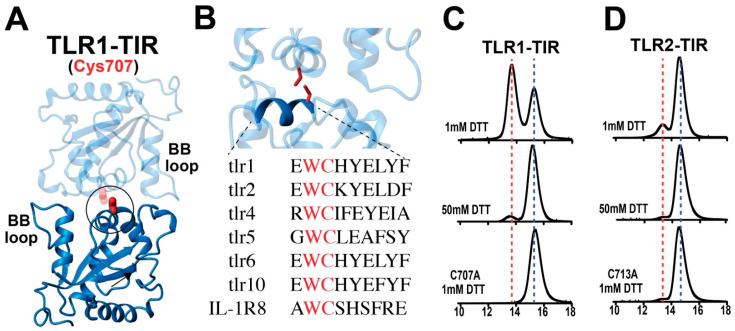
TLR1-TIR and TLR2-TIR homodimerization. (**A**) A conserved cysteine, C707 (red), within the crystallographic homodimer in TLR1 is shown (PDB accession 1fyv). (**B**) The cysteine is partially buried by the conserved BB-loop and is conserved within multiple TLR TIR domains and the IL-1R8-TIR domain. (**C**) Analytical Superose-12 of recombinant TLR1-TIR in 1 mM DTT (top), incubated for 30 min in 50 mM DTT (middle), and the TLR1-TIR C707A mutant in 1 mM DTT (bottom). (**D**) Analytical Superose-12 of recombinant TLR2-TIR in 1 mM DTT (top), incubated for 30 min in 50 mM DTT (middle), and the TLR2-TIR C713A mutant in 1 mM DTT (bottom). Buffer for analytical sizing was 50 mM Na_2_HPO_4_, pH 6.5, 150 mM NaCl, 1 mM DTT.

**Figure 6 molecules-27-04494-f006:**
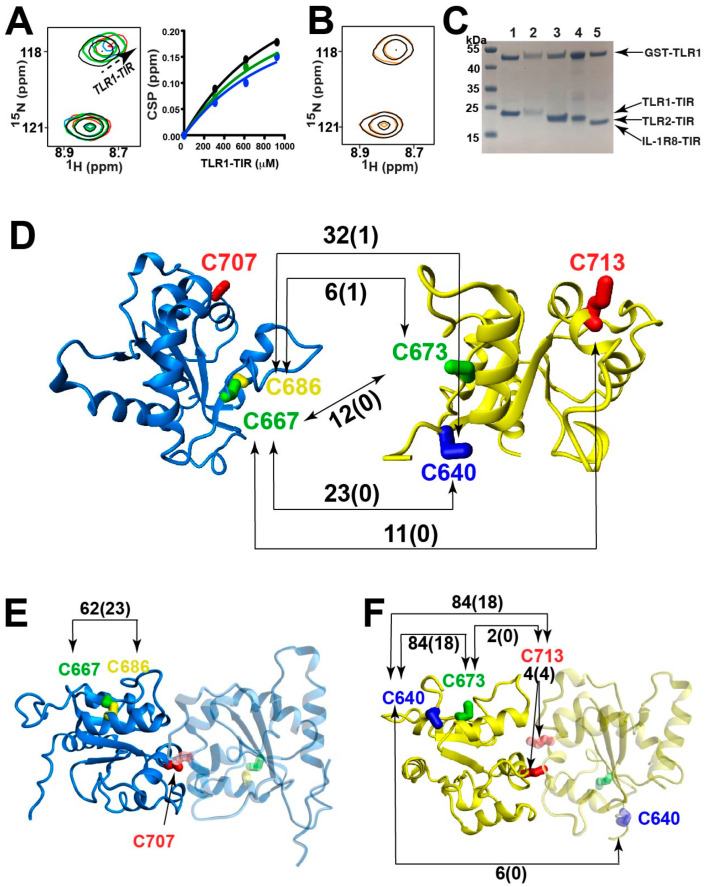
TLR1-TIR and TLR2-TIR heterodimerization. (**A**) ^15^N-TLR2-TIR at 308 μM (black) was titrated with TLR1-TIR at 308 μM (green), 616 μM (blue), and 924 μM (red), with the associated binding isotherm derived from several resonances shown and their global fit resulting in 760 ± 230 μM. (**B**) ^15^N-TLR2-TIR at 308 μM (black) was titrated with full-length tirA at 308 μM (orange). (**C**) Disulfide-mediated cross-linking, pull-down experiments using GST-TLR1 were performed with untagged TLR1-TIR (lane 1), the TLR1-TIR C707A mutant (lane 2), TLR2-TIR (lane 3), the TLR2-TIR C713A mutant (lane 4), and IL1-R8 TIR (lane 5). (**D**) MS identified disulfide-linked peptides between TLR1-TIR and TLR2-TIR, with the number of peptides identified prior to treatment shown above the arrows and after reduction within parenthesis. (**E**) MS identified disulfide-linked peptides for TLR1-TIR, with the number of peptides shown. (**F**) MS identified disulfide-linked peptides for TLR2-TIR, with the number of peptides shown.

**Figure 8 molecules-27-04494-f008:**
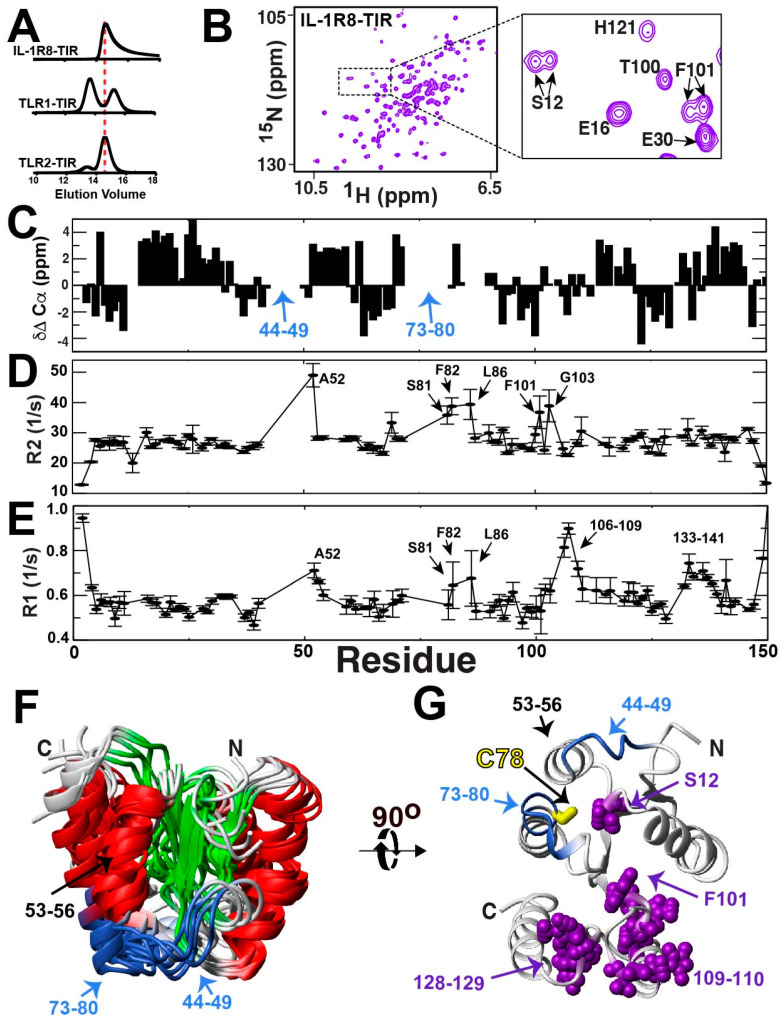
IL-1R8-TIR domain solution studies. (**A**) Size-exclusion analysis of recombinant IL-1R8-TIR (residues 160–310, renumbered as 1–150) using an analytical Superose-12 (23.4 mL). (**B**) 15N-HSQC of the IL-1R8-TIR domain, with the inset indicating multiple resonances for both S12 and F101. (**C**) CA chemical shift differences relative to a random coil (δΔ CA), which provides the propensities for secondary structure. (**D**) R2 relaxation rates for the IL-1R8-TIR domain. (**E**) R1 relaxation rates for the IL-1R8-TIR domain. (**F**) Superposition of the top five structures determined from CS-Rosetta with an RMSD of 2.04 Å. From chemical shift data, CSI version 2.0 [36] was used to calculate secondary structure mapped as β-strand (green) for residues 4–12, 36–40, 63–68,95–102, and 123–128 and as a-helices (red) for residues 15–32, 51–60, 81–90, 111–120, and 138–146. Additionally, the missing resonances of 44–49 and 73–80 are shown (blue), and the position of the BB-loop, i.e., residues 53–56, forming an α-helix, is shown. (**G**) A 90° rotation of one calculated structure is used to indicate how the conserved C78 is pointing inward along with residues exhibiting multiple resonances (purple bonds), which include S12, F101, G103, A109, H110, R120, W128, and R129.

## Data Availability

Chemical shift assignments for the tirE-CC and IL-1R8-TIR have been deposited in the BMRB (http://www.bmrb.wisc.edu/, accessed on 20 May 2022) with associated accession numbers 51474 and 51473, respectively.

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
