# Peer review of "Human and Bacterial Toll-Interleukin Receptor Domains Exhibit Distinct Dynamic Features and Functions"

_molecules, 2022, doi:10.3390/molecules27144494_

Round 1
Reviewer 1 Report
In the manuscript Eisenmesser an co-workers has made an extensive characterisation on the structure, function and dynamics of both mammalian and bacterial TIR domains. The study is a combined effort centred on NMR spectroscopy and complemented with mass spectrometry, quantitative size exclusion chromatography and structural modelling, For the bacterial TIR domains the authors demonstrate a catalytic activity of the proteins towards NAD+. A real-time NMR approach was developed that convincingly demonstrated the break-down of NAD+ into ADP-ribose and Nicotinamide. For the bacterial TIR domains the authors further pinpointed the dimerization function to the so called “CC” domain. A model for the secondary structure composition of the CC domain of tirE was put forward and this is relevant since no structural data is available for this family of oligomerisation domains. For the mammalian TIR domains the authors demonstrated a redox dependent dimerization of TIR domains that is effectuated through disulphide linkages. Finally a csROSETTA derived structural model was made for the mammalian IL-1R8 TIR domain. In summary the authors has laid a solid foundation for the structure and function of both bacterial and mammalian TIR domains. The manuscript is comprehensive and well written and the experiments and data analysis are well performed. Except for some minor requests (as listed below) my recommendation is that the manuscript is accepted for publication in “Molecules”.
Minor points:
1) Figure 4D would benefit from the secondary structure prediction on top of the diagram.
2) The discussion on the role of disulphide-dimers can be expanded to include a more in depth discussion on cellular consequences of homo- and hetero-dimer formation.
3) In figure 6 I suspect that “microM” has been replaced with “milliM” on a number of instances.
4) The structural model in Figure 8F deserves a more detailed analysis, does there exist charged or hydrophobic patches that may serve as protein-protein interaction hot-spots? Also a comparison to other related structures is suggested.
5) Figure S3 is cropped and should be corrected
Author Response
1) Figure 4D would benefit from the secondary structure prediction on top of the diagram.
We have placed the secondary structural assignments as predicted by CSI within Figure 4D to match the cartoon representation of helices (depicted as blue barrels) along with the structural prediction of the helices from CS-Rosetta as suggested by Reviewer 2. We thank the reviewer for this suggestion to help clarify the secondary structure.
2) The discussion on the role of disulphide-dimers can be expanded to include a more in-depth discussion on cellular consequences of homo- and hetero-dimer formation.
We have now expanded on both the anti-inflammatory and pro-inflammatory roles that cysteine-mediated interactions may play within the Discussion regarding their potential regulation of signaling.
3) In figure 6 I suspect that “microM” has been replaced with “milliM” on a number of instances.
Yes, you are correct and thank you.
4) The structural model in Figure 8F deserves a more detailed analysis, does there exist charged or hydrophobic patches that may serve as protein-protein interaction hot-spots? Also a comparison to other related structures is suggested.
We have now included an additional Supplementary Figure 6 that compares the IL-1R8-TIR model to the closest structural homologue, which is the MAL adaptor protein. Regarding charge distributions, most TIR domains attached to mammalian receptors are relatively basic, which makes it difficult to understand solely from their structures how they may form heterodimers or even larger complexes. IL-1R8-TIR is highly basic (pI~10) and thus, it does indeed have multiple basic patches. However, MAL has been shown to form filaments that are largely dictated by main chain atoms with adjustments to flexible regions with substantial changes in internal disulfide bonds that IL-1R8-TIR does not comprise. This likely means that IL-1R8-TIR could undergo conformational changes even more readily than MAL, but deducing its signaling complex still requires trapping these experimental complexes likely through similar cryo-EM studies.
5) Figure S3 is cropped and should be corrected.
Yes, you are correct and thank you.
Reviewer 2 Report
In this manuscript, Eisenmesser and co-workers investigated the dynamic behavior of bacterial TIR-domain containing proteins and mammalian TIR domains using biochemical and biophysical methods. Using NMR they found that bacterial TIR domains undergo a global exchange on the intermediate timescale while the mammalian TIR domains exhibited a range of dynamic exchange spanning multiple timescales. Only the bacterial TIR domains were able to cleave NAD+. The authors also describe differences in oligomerization. Bacterial TIR-domain containing proteins formed dimers through a putative coiled-coil domain, while mammalian TIR domains formed homo-and heterodimers via a dock and lock mechanism involving cysteine residues.
To me, the manuscript seems like a collection of many rather preliminary results. For example, I found the investigation on the dimerization domain interesting, but was disappointed that it was not continued because NMR and X-ray crystallography did not work for structure determination. I was expecting that the authors would use at least AlphaFold to predict the structure of the dimerization domain and compare it to the NMR results. Certainly, such a comparison would have led to a hypothesis that could have been further tested. Likewise, I am not convinced that the TLR1- TIR/TLR2-TIR Cys-crosslinking results in specific heterodimers, but unfortunately, there are no additional results to confirm this. Furthermore, I also wonder if the purification of the proteins under denaturing conditions could have an impact on the quality of the samples.
There are many typos throughout the manuscript.
Author Response
1) I found the investigation on the dimerization domain interesting, but was disappointed that it was not continued because NMR and X-ray crystallography did not work for structure determination. I was expecting that the authors would use at least AlphaFold to predict the structure of the dimerization domain and compare it to the NMR results.
Although there is no AlphaFold prediction for tirE, most predictions that include the AlphaFold prediction of tirS comprise helices for the majority of the CC domain, as they are likely based on structures from other species that share little sequence similarity. Thus, to address the reviewer’s point, we have now included the CS-Rosetta derived predictions of the first two helices and discussions regarding this experimentally driven model structure in Figure 4F.
2) Certainly, such a comparison would have led to a hypothesis that could have been further tested.
We do agree that further testing the specific residues within the tirE-CC domain could potentially lead to the identification of the explicit residues responsible for dimerization (i.e., alanine scanning or swaps of charged residues). However, there is no guarantee that such mutations would be successful and the major point of these investigations was to determine the region of the bacterial TIR-containing domains responsible for dimerization, which is the CC domain. We anticipate that the full structural elucidations of these domains will be determined by the successful characterization of such CC regions within the context of larger oligomeric complexes. Unfortunately, trapping such complexes has been difficult through the “purest approach” that comprises independently purifying different proteins thought to associate, as it is likely other protein components are simply missing. We have desperately tried to pull-down such complexes from cells transfected with the bacterial TIR-containing domains as well as the mammalian TIR domains, but have yet to be successful.
3) Likewise, I am not convinced that the TLR1- TIR/TLR2-TIR Cys-crosslinking results in specific heterodimers, but unfortunately, there are no additional results to confirm this. Furthermore, I also wonder if the purification of the proteins under denaturing conditions could have an impact on the quality of the samples.
We apologize for any ambiguity in the purification protocols, as refolded TLR1-TIR and TLR2-TIR domains do match the soluble forms that simply had led to far less protein product than the refolds of these domains. Moreover, the 15N-HSQC of TLR1-TIR and TLR2-TIR domains match spectra published in Lushpa et al. (2021) and Nada et al. (2012), which we have specifically described in Materials and Methods now. Although there are no spectra previously published for the IL-1R8-TIR domain and the bacterial domains, it seems unlikely that IL-1R8-TIR is misfolded considering that the structural elucidations reveal a TIR-like domain and the bacterial TIR domains were all active enzymes. Thus, refolding does not appear to be a problem here.
Reviewer 3 Report
Dear Authors,
My congratulations for a good work. I would like to propose only several minor comments:
1. p9, line 299. The phrase “These point mutations are still well-folded …” seems to be incorrect because the protein mutants are still well-folded.
2. p9, Figures 5c,d. It would be useful to clarify in the caption the buffer by which the column was equilibrated, and why at 1 mM DTT the samples are not incubated.
3. p11, line 365. The protein concentration is shown in mM, but at Figure 6a itself – in µM.
4. p11, lines 376-377. Indicate, please, what are reducing conditions.
Author Response
1) p9, line 299. The phrase “These point mutations are still well-folded …” seems to be incorrect because the protein mutants are still well-folded.
Apologies, as we simply meant to say that “These point mutations are well-folded”.
2) p9, Figures 5c,d. It would be useful to clarify in the caption the buffer by which the column was equilibrated, and why at 1 mM DTT the samples are not incubated.
This is a very good point of clarification that we have now added to this figure, as the buffer already had 1 mM DTT. We have also specifically described the use of 1 mM DTT for the mammalian proteins in Materials and Methods that was supplemented during denaturation, refolding, and final purification. We apologize for neglecting this important detail.
.3) p11, line 365. The protein concentration is shown in mM, but at Figure 6a itself – in µM.
Apologies, as these were all supposed to be in micromolar and they have now been changed. Thank you.
4) p11, lines 376-377. Indicate, please, what are reducing conditions.
Our implication here was to indicate that disulfides are formed under “reducing conditions” that are expected within the cytosol of the cell, but we were unclear in this. We have now specifically stated “….under reducing conditions such as those expected within the environment of a cell” in order to be specific. We apologize for not stating this more concretely before.
Round 2
Reviewer 2 Report
In my view, the modifications didn't improve the manuscript. The part about the dimerization domains is even more confusing than before. Rather than trying to discuss away the points raised, the authors should perform the appropriate experiments.
1) Actually, the suggestion was to predict the structure of the dimeric dimerization domain, to obtain possible insights on how the dimer is formed. It is not clear to me what we can learn from the monomeric model shown in Fig. 4 about dimerization.
"Although there is no AlphaFold prediction for tirE..." That is exactly the point. The authors could have used AlphaFold to predict the tirE dimer...
2) "102-146 do not sample an a-helical structure, but instead exhibit a beta-strand-like structure that is predicted to be a random coil by CSI version 2.0 [36] (Figure 4D)." "This weak beta-strand propensity predicted to be a coil-coil structure for residues 102-146...". Is it now predicted to be a random coil or a coiled coil? I find this very confusing. Which part is forming the dimer (1-102 or 102-146)? That could have been easily tested. With a strong AlphaFold model of the dimer there would be no need for doing an Ala scan for identifying putative regions for dimerization.
3) The point concerning the "specificity" of disulfide-bond formation between TLR1-TIR and TLR2-TIR was not addressed.
Author Response
Reviewer#2: In my view, the modifications didn't improve the manuscript. The part about the dimerization domains is even more confusing than before. Rather than trying to discuss away the points raised, the authors should perform the appropriate experiments.
Response: We did not mean to “discuss away”, as the request for an AlphaFold prediction was simply not possible considering that AlphaFold could not even predict a monomer and that is why we used a Rosetta-based method to incorporate experimental data instead (described below for our particular case). The limitations of AlphaFold have been described by many experts in the field, which include the recent description by Moore, Hendrickson, Henderson, and Brunger (“The protein-folding problem: Not yet solved”. Science DOI: 10.1126/science.abn9422). While we could have reported the failure of AlphaFold along with the success of Rosetta-based predictions provided in our previously revised manuscript, we did not immediately believe that this was necessary for our manuscript when your request was to provide a model based on our NMR data. However, we apologize for this lack of clarity and have now included these findings within our explanation within the main text of the Results (see yellow highlight).
Reviewer#2: 1) Actually, the suggestion was to predict the structure of the dimeric dimerization domain, to obtain possible insights on how the dimer is formed. It is not clear to me what we can learn from the monomeric model shown in Fig. 4 about dimerization. "Although there is no AlphaFold prediction for tirE..." That is exactly the point. The authors could have used AlphaFold to predict the tirE dimer...
Response: Specifically, AlphaFold did NOT predict any structure for the tirE-CC domain, which can be found here : https://www.uniprot.org/uniprotkb/A0A097BYB4/entry. This is why we instead utilized Chemical Shift (CS)-Rosetta that incorporated the experimental chemical shifts and did converge on the predicted alpha-helical structure for the first ~100 amino acids, which is the first of any experimentally driven atomic description. Although we have shown experimentally that this CC domain is the dimerization domain (Figure 3) and now provided the monomeric structure from our NMR chemical shifts (Figure 4), the reviewer is correct in that we still do not know explicitly how the dimer is formed. This is not from lack of trying through biophysical methods. Thus, our current strategy has turned to utilizing the full-length bacterial TIR proteins (that include the CC domains) for cellular pull-down assays in order to apply cryo-EM studies to larger complexes and thereby directly identify the structure of the CC dimer. However, thus far such pull-downs have not been successful using several model cell lines that include both lung cell likes (A549 cells) and cell lines of hematopoietic origin (THP-1 cells), which are known to comprise multiple TLR receptors that could be targeted by these bacterial TIR containing proteins.
Reviewer#2: 2) "102-146 do not sample an a-helical structure, but instead exhibit a beta-strand-like structure that is predicted to be a random coil by CSI version 2.0 [36] (Figure 4D)." "This weak beta-strand propensity predicted to be a coil-coil structure for residues 102-146...". Is it now predicted to be a random coil or a coiled coil? I find this very confusing. Which part is forming the dimer (1-102 or 102-146)? That could have been easily tested. With a strong AlphaFold model of the dimer there would be no need for doing an Ala scan for identifying putative regions for dimerization.
Response: We apologize for this lack of clarity, as we had meant “This weak beta-strand propensity THAT WAS INITIALLY PREDICTED TO BE A HELIX WITHIN A COIL-COIL”, so we have rewritten this statement in the main text to make this clearer (see yellow highlight). As stated above, structural calculations of the dimer are not “easily tested”, as no “strong AlphaFold model” exists for even a monomer much less a dimer, which is why this study is so important in providing the first experimental data on this region that exists within multiple bacterial TIR domains.
Reviewer#2: 3) The point concerning the "specificity" of disulfide-bond formation between TLR1-TIR and TLR2-TIR was not addressed.
Response: The reviewer brings up an important point that we have now tried to clarify, as the mass spectrometry analysis indicates that most of the cysteine residues can form heterodimers and thus, there is no apparent specificity for heterodimerization. This is in complete contrast to homodimizeration, which we show IS SPECIFIC for a single conserved cysteine residue within both TLR1-TIR and TLR2-TIR (Figure 5). Such complexities that could indicate heterogeneity for cellular signaling could also explain the complete lack of structural data for these complexes, as homogeneous samples are difficult to produce when instead, these interactions comprise heterogeneous mixtures. We apologize for not making this clearer initially and have rewritten these findings in relation to heterodimer formation in both the Results and Discussion (see yellow highlights).